# Tight bending of the Ndc80 complex provides intrinsic regulation of its binding to microtubules

Emily Anne Scarborough[1], Trisha N Davis[1]*, Charles L Asbury[2]*

[1]Department of Biochemistry, University of Washington, Seattle, United States; [2]Department of Physiology and Biophysics, University of Washington, Seattle, United States

**Abstract** Regulation of the outer kinetochore complex Ndc80 is essential to ensure correct kinetochore-microtubule attachments during mitosis. Here, we present a novel mechanism of regulation that is intrinsic to its structure; tight bending of the Ndc80 complex inhibits its microtubule binding. Using single molecule Förster resonance energy transfer (FRET), we show that the *Saccharomyces cerevisiae* Ndc80 complex can fluctuate between straight and bent forms, and that binding of the complex to microtubules selects for straightened forms. The loop region of the complex enables its bent conformation, as deletion of the loop promotes straightening. In addition, the kinetochore complex MIND enhances microtubule binding by opposing the tightly bent, auto-inhibited conformation of the Ndc80 complex. We suggest that prior to its assembly at the kinetochore, the Ndc80 complex interchanges between bent (auto-inhibited) and open conformations. Once assembled, its association with MIND stabilizes the Ndc80 complex in a straightened form for higher affinity microtubule binding.
DOI: https://doi.org/10.7554/eLife.44489.001

*For correspondence:
tdavis@uw.edu (TND);
casbury@uw.edu (CLA)

Competing interests: The authors declare that no competing interests exist.

## Introduction

The highly conserved kinetochore complex Ndc80 (*Lampert et al., 2013*; *Tien et al., 2013*; *Umbreit et al., 2012*; *McCleland et al., 2003*) is a hub of function within the kinetochore. It is directly involved in kinetochore-microtubule attachment, in bearing mechanical load at the kinetochore-microtubule interface, and in correction of erroneous attachments (*Powers et al., 2009*; *Umbreit et al., 2012*; *Etemad et al., 2015*). The Ndc80 complex specifically localizes only to kinetochores and binds the plus-ends of microtubules in vivo. Even in instances of Ndc80 overexpression in yeast or mammalian cells, excess Ndc80 is not found along the microtubule lattice in the mitotic spindle (*Tang and Toda, 2015*; *Diaz-Rodríguez et al., 2008*). This specific localization to the tips of microtubules mirrors the localization of +TIPs, which often require multiple modes of regulation to ensure accurate targeting to the microtubule plus-end as well as appropriate function (*Gireesh et al., 2018*). Thus, there are likely multiple regulatory mechanisms to ensure both proper localization and function of the Ndc80 complex in cells.

The Ndc80 complex is an elongated and flexible heterotetramer, with two globular ends connected by a coiled coil, which displays a variety of conformations in negative stain electron micrographs (*Wang et al., 2008*; *Zhang et al., 2012*) and in crystallized forms (*Ciferri et al., 2008*; *Valverde et al., 2016*). Studies suggest that its flexibility is important in vivo (*Wang et al., 2008*; *Maure et al., 2011*; *Joglekar et al., 2009*; *Tien et al., 2014*). For example, the ability of the Ndc80 complex to adopt both straightened and bent conformations is necessary for the proper progression of mitosis and implicated in the resolution of aberrant kinetochore-microtubule attachments (*Tien et al., 2014*). Cross-linking data suggest that this flexibility is extreme in its range: the Ndc80

complex not only bends, but also has the capacity to adopt a tightly bent conformation (*Tien et al., 2014*). However, the effect that these conformational changes have on the Ndc80 complex's microtubule-binding activity is unknown. Several studies have suggested that microtubule binding by the Ndc80 complex is auto-inhibited (*Cheeseman et al., 2006*; *Kudalkar et al., 2015*). But it remains unclear whether the underlying mechanism of auto-inhibition depends on conformational changes of the complex, or how such regulation might influence Ndc80 complex function in vivo.

The MIND/Mtw1 complex bridges the inner and outer layers of the kinetochore, directly binding the Ndc80 complex and tethering it to the DNA-binding kinetochore components (*Maskell et al., 2010*). In vitro, the addition of MIND to the Ndc80 complex increases its residence time on the microtubule lattice by nearly fourfold (*Kudalkar et al., 2015*). Based on this and other observations, we previously proposed an allosteric mechanism by which the Ndc80 complex interchanges between a more open form when bound to the MIND complex, and a bent, auto-inhibited state with decreased microtubule binding when unbound from MIND (*Kudalkar et al., 2015*).

Direct evidence implicating tight bending in the auto-inhibition of Ndc80 complex is lacking, as no study has yet examined conformational changes of the complex while simultaneously monitoring its microtubule-binding activity. To do so, we used single molecule techniques including total internal reflection fluorescence (TIRF) microscopy and single molecule FRET. We found that the *Saccharomyces cerevisiae* Ndc80 complex is auto-inhibited, can tightly bend, and that the addition of MIND opposes the tightly bent conformation. The ability of the Ndc80 complex to change conformations, and thus tune its microtubule affinity, might play an important role for proper localization within the cell. Our work reveals a previously unstudied mode of inherent regulation at the kinetochore.

## Results

### Microtubule binding by the Ndc80 complex is auto-inhibited by its Spc24/Spc25 end

Previous work (*Kudalkar et al., 2015*; *Cheeseman et al., 2006*) suggested the possibility of an intra-molecular, auto-inhibitory interaction in which the tetrameric Ndc80 complex bends tightly, allowing its Spc24/Spc25 end to associate closely with, and reduce the microtubule-binding of its Ndc80/Nuf2 end (*Figure 1A*). To test directly for inhibition of Ndc80/Nuf2 by Spc24/Spc25, we measured the microtubule binding of a shortened dimeric construct, called 'Broccoli' (*Schmidt et al., 2012*) consisting of just the Ndc80 and Nuf2 proteins truncated to remove their C-termini (Ndc80 1–556, Nuf2 1–339), through which they would normally tetramerize with Spc24 and Spc25 (*Wei et al., 2005*). A yeast Broccoli construct with GFP fused C-terminally to Nuf2 was purified (Broccoli-GFP; *Figure 1—figure supplement 1A*) and its binding to taxol-stabilized microtubules was observed using single molecule total internal reflection fluorescence (TIRF) microscopy. Full-length, GFP-tagged tetrameric Ndc80 complex was also examined as a control. Consistent with previous studies (*Kudalkar et al., 2015*; *Powers et al., 2009*), individual full-length Ndc80 complexes bound transiently and relatively weakly to microtubules, with a mean residence time of 2.9 ± 0.3 s (*Figure 1B and C*). Individual Broccoli-GFP dimers bound longer, with a mean residence time of 5.6 ± 0.7 s, roughly twice as long as full-length tetrameric Ndc80-GFP, suggesting that removal of the Spc24/Spc25 end of the complex relieves an auto-inhibitory effect (*Figure 1B and C*).

We reasoned that if the Spc24/Spc25 end is responsible for auto-inhibition within the full-length complex, then free Spc24/Spc25 dimer added in trans might be sufficient to inhibit microtubule binding. Indeed, when free Spc24/Spc25 dimer was purified (*Figure 1—figure supplement 1C*) and added in sixtyfold excess to Broccoli-GFP, the mean residence time of Broccoli-GFP on microtubules was reduced to 3.3 ± 0.4 s, which is statistically indistinguishable (p=0.4) from the residence time of full-length Ndc80 complex alone. Adding MIND-GFP to full-length Ndc80 complex increased its residence time on microtubules more than threefold, from 2.9 ± 0.3 s to 9.8 ± 1.1 s (*Figure 1B and C*), consistent with previous measurements (*Kudalkar et al., 2015*). This MIND-dependent increase was almost completely abolished by addition of a tenfold excess of Spc24/Spc25 dimer, which brought the residence time for the Ndc80-MIND-GFP co-complex down to 4.1 ± 0.5 s (*Figure 1B and C*), similar to that of full-length tetrameric Ndc80 alone. Likewise, a mutant Spc24/Spc25 dimer that is unable to associate with MIND (*Malvezzi et al., 2013*), when added in tenfold excess, also sharply reduced the residence time of the Ndc80-MIND-GFP co-complex, down to 4.2 ± 0.5 s (*Figure 1—*

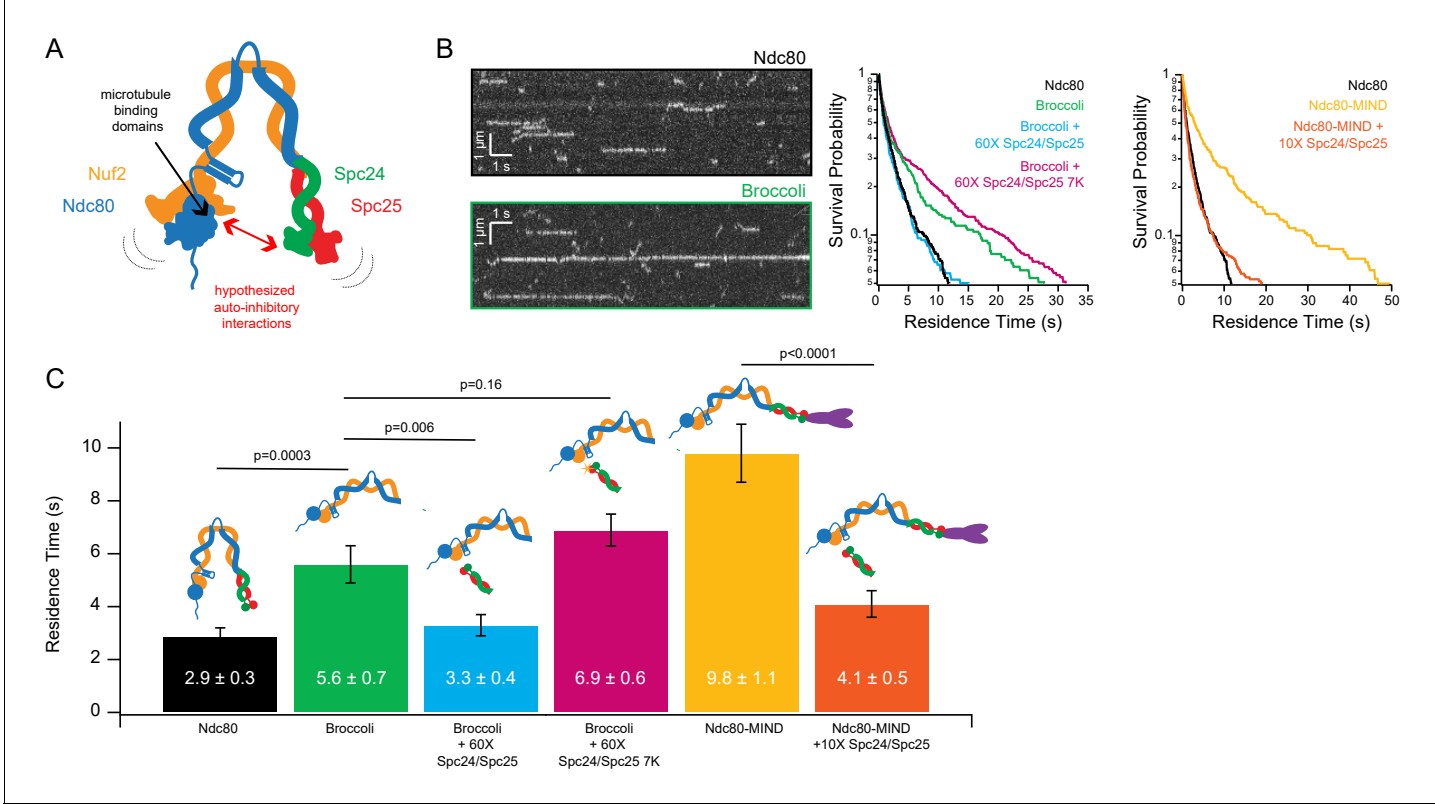

**Figure 1.** Ndc80 complex microtubule binding is auto-inhibited by the Spc24/Spc25 dimer. (A) Cartoon of hypothesized model of auto-inhibition of the Ndc80 complex. Red arrow indicates hypothesized regions of intra-complex interactions. Black arrow indicates calponin homology domains on the Ndc80/Nuf2 dimer responsible for microtubule binding. (B) (*Left*) Representative kymographs of Ndc80 and Broccoli. (*Right*) Survival probability curves of residence times of the Ndc80 complex (n = 537), Broccoli (n = 315, Broccoli + 60X Spc24/Spc25 (n = 398), Broccoli + 60 x Spc24/Spc25 7K (n = 586) and the Ndc80 complex (repeated), Ndc80-MIND (n = 279), Ndc80-MIND + 10X Spc24/Spc25 (n = 674). (C) Bar graph of average residence times of the data represented in survival probability curves in *Figure 1B*. Error was calculated using bootstrapping analysis. p-Values were calculated using a two-tailed Student's *t* test. Raw data of all residence times are included in *Figure 1—source data 1*. Additional supplementary data are included in *Figure 1—figure supplement 1*, *Figure 1—figure supplement 2* and *Figure 1—figure supplement 3*.

DOI: https://doi.org/10.7554/eLife.44489.002

The following source data and figure supplements are available for figure 1:

**Source data 1.** Residence times on microtubules measured for individual complexes of Ndc80, Broccoli, and Ndc80-MIND co-complexes, without and with the addition of Spc24/Spc25 dimer.

DOI: https://doi.org/10.7554/eLife.44489.008

**Figure supplement 1.** Purifications of Ndc80 complex constructs.

DOI: https://doi.org/10.7554/eLife.44489.003

**Figure supplement 1—source data 1.** Residences times on microtubules measured for Ndc80 complexes, without and with the addition of a large excess of Spc24/Spc25 dimer.

DOI: https://doi.org/10.7554/eLife.44489.004

**Figure supplement 2.** Representative kymographs of TIRF microscopy assays.

DOI: https://doi.org/10.7554/eLife.44489.005

**Figure supplement 3.** Spc24/Spc25 V159D dimer can inhibit the Ndc80-MIND co-complex.

DOI: https://doi.org/10.7554/eLife.44489.006

**Figure supplement 3—source data 1.** Residence times on microtubules measured for individual Ndc80-MIND co-complexes, without and with the addition of mutant Spc24/Spc25 V159D dimer.

DOI: https://doi.org/10.7554/eLife.44489.007

*figure supplement 3*). This observation indicates that the excess Spc24/Spc25 dimer interacts directly with the Ndc80 complex for inhibition, rather than competing away MIND.

We also tested whether full-length Ndc80 complex without MIND could be further inhibited by excess free Spc24/Spc25 dimer. For this test, we increased the dynamic range of our measurements

by replacing our standard BRB80 buffer with BRB60, which lowers the concentration of cations and thereby raises the mean residence time for full-length Ndc80 complex up to 5.7 ± 0.7 s (*Figure 1—figure supplement 1B*). Addition of free Spc24/Spc25 dimer under these conditions decreased the mean residence time of Ndc80-GFP complex in a concentration-dependent manner (*Figure 1—figure supplement 1B*). Free Spc24/Spc25 dimer does not detectably bind microtubules by itself (*Wei et al., 2007*). Altogether, our observations show that the Spc24/Spc25 dimer can interact with Ndc80/Nuf2 independently of the tetramerization domain, and that this secondary interaction is sufficient to inhibit the microtubule binding of Ndc80/Nuf2, of the full-length Ndc80 complex, and of the MIND-Ndc80 co-complex.

## Acidic residues in Spc25 are important for auto-inhibition

Electrostatic interactions are crucial for binding of the Ndc80 complex to microtubules. Basic residues located on the globular (calponin homology) domains of the Ndc80 and Nuf2 subunits have been shown to interact with acidic patches on the surface of the microtubule to enable microtubule binding (*Ciferri et al., 2008*). Additionally, charge-charge interactions between the N-terminal 'tail' of Ndc80 and the microtubule lattice tune this microtubule binding through phosphorylation (*Zaytsev et al., 2015*; *Zaytsev et al., 2014*). Considering the central role of electrostatic interactions in microtubule-binding of the Ndc80 complex, we hypothesized that similar electrostatic effects might also contribute to the intra-complex interactions underlying its auto-inhibition. Sequence alignment revealed three acidic residues located on one face of the Spc25 subunit that are highly conserved, suggesting functional importance (*Figure 1—figure supplement 1E*). To probe whether negative charges on this face of Spc25 contribute to inhibitory interactions, we mutated a total of seven acidic residues – including the three identified by sequence conservation plus four others located nearby on the same face – to lysine residues. We then purified a charge-reversal mutant dimer of Spc24/Spc25 carrying the lysine substitutions (hereafter referred to as Spc24/Spc25-7K) (*Figure 1—figure supplement 1C and D*). When added in sixtyfold excess, Spc24/Spc25-7K failed to detectably inhibit the microtubule-binding activity of Broccoli-GFP (mean residence time of 6.9 ± 0.6 s with versus 5.6 ± 0.7 s without the addition Spc24/Spc25-7K; p=0.16), in contrast to the wild-type Spc24/Spc25 dimer (3.3 ± 0.4 s with versus 5.6 ± 0.7 s without; p=0.006) (*Figure 1B and C*). While a full-length, tetrameric Ndc80 complex carrying the 7K charge-reversal mutations could not be purified (see Materials and methods), the normal microtubule-binding activity of the Ndc80 complex in the presence of a large excess of Spc24/Spc25-7K dimer nevertheless suggests that electrostatic interactions play an important role in auto-inhibition of the complex.

## The Ndc80 complex can bend tightly

Electron micrographs of purified, negatively stained samples of Ndc80 complex show that the coiled-coil can adopt a variety of conformations, with a tendency to bend at or near the 'loop,' an unstructured region of 50–60 residues within the Ndc80 protein (*Wang et al., 2008*; *Zhang et al., 2012*). Genetic evidence suggests the complex must bend tightly to function properly in vivo (*Tien et al., 2014*), but the propensity of the complex for tight bending has not been directly examined. To monitor bending of the Ndc80 complex, we used Förster resonance energy transfer (FRET). We first created plasmids encoding a cysteine-free version of the complex by mutating all eight of its native cysteine codons to serine codons. Then two new cysteines were introduced, replacing native surface-exposed serines at either end of the complex, near the N-terminus of Nuf2 (S2C) and on the RWD domain of Spc25 (S154C) (*Figure 2A and B*). Sequential labeling with cysteine-reactive dyes during purification created a homogenous population of complexes, each with a Cy3 dye on the Spc24/Spc25 end and a Cy5 dye on the Ndc80/Nuf2 end, which bound to microtubules with an affinity similar to that of Ndc80-GFP (*Figure 2C*, and *Figure 2—figure supplement 1A, B and C*). At these locations, the two dyes should not come into close enough proximity for FRET (i.e. less than 10 nm apart) unless the complex adopts a tightly bent configuration (*Figure 2D*).

To test for tight bending of the Ndc80 complex in solution, we began by measuring the end-labeled complex in a bulk FRET assay (*Figure 2E*). For comparison, we also examined positive and negative controls made from two single-stranded DNA oligonucleotides labeled with Cy3 and Cy5. For the positive control, the labeled oligos were annealed to a target strand, creating a double-stranded DNA assembly with the dyes held in close proximity (*Friedman et al., 2006*). For the

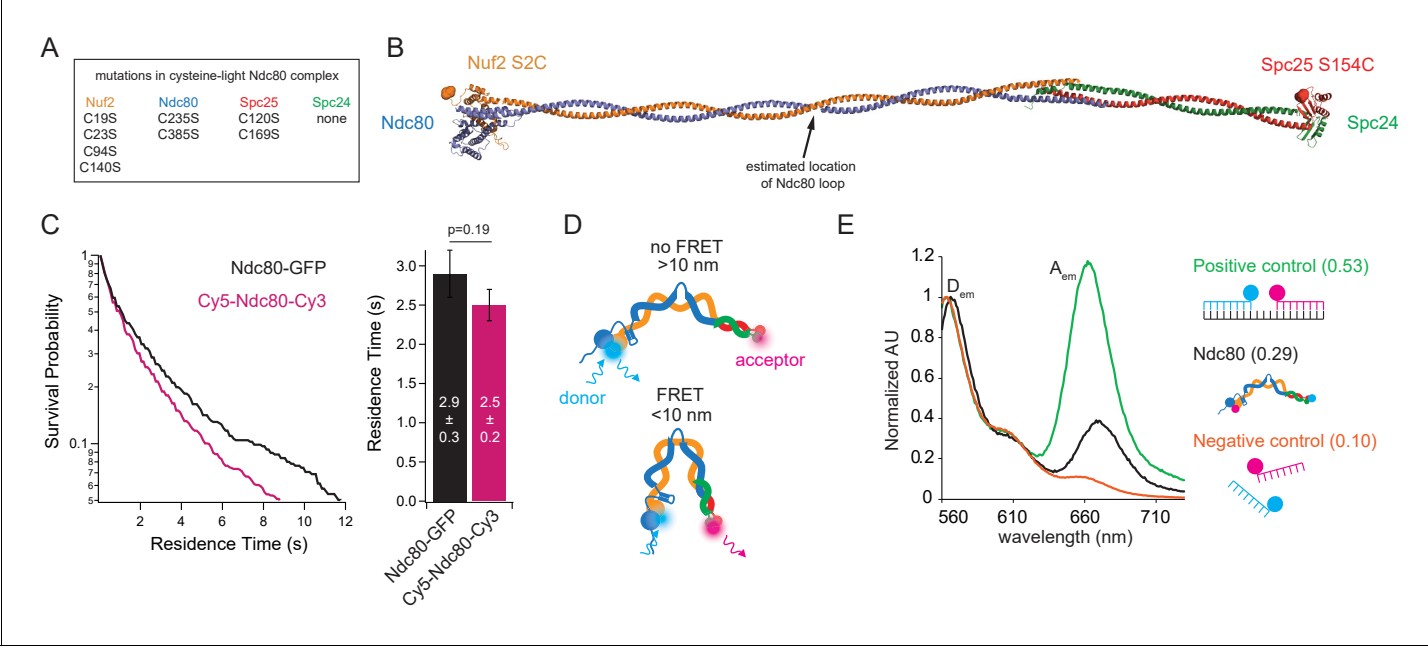

**Figure 2.** The Ndc80 complex can tightly bend. (**A**) List of mutations made in the wild type Ndc80 complex to generate cysteine-light construct. (**B**) Model of the Ndc80 complex using the dwarf tetramer structure (DOI: 10.2210/pdb5TCS/pdb) plus inserted coiled coil between the tetramerization domain and globular domain of each dimer. Break in coiled coil indicates estimated location of Ndc80 loop, indicated by black arrow. Colored balls on each globular domain of the model indicate location of the cysteine residues (Nuf2 S2C, Spc25 S154C) used for dual end-labeling. (**C**) (*Left*) Survival probability curves of residence times for Ndc80-GFP (data repeated from **Figure 1B**) (black trace) and end-labeled Cy5-Ndc80-Cy3 (n = 1375) (pink trace). (*Right*) Bar graph of average residence times of the data at left. Error was calculated using bootstrapping analysis. p-Value was calculated using a two-tailed Student's *t* test. Raw data of all residence times are included in **Figure 2—source data 1**. (**D**) Cartoon of approximate dye locations on the Ndc80 complex illustrating that in a more open conformation (*Top*), FRET would not occur. In a tightly bent conformation (*Bottom*), the dyes would be in close proximity and exhibit FRET. (**E**) Bulk FRET spectra of positive control double-stranded, dually-labeled oligonucleotide, negative control single-stranded singly-labeled oligonucleotides or end-labeled Ndc80 complex Nuf2 S2C Spc25 S154C. Displayed is the emission spectra of each construct from 560 nm to 730 nm under 550 nm (Cy3, donor) excitation. $D_{em}$ and $A_{em}$ denote the emission peaks of the donor (Cy3) and acceptor (Cy5). Values in parentheses indicate calculated relative FRET efficiency. Additional supplementary data are included in **Figure 2—figure supplement 1**.

DOI: https://doi.org/10.7554/eLife.44489.009

The following source data and figure supplement are available for figure 2:

**Source data 1.** Residence times on microtubules measured for individual, dual end-labeled Ndc80 complexes.

DOI: https://doi.org/10.7554/eLife.44489.011

**Figure supplement 1.** A functional Ndc80 complex can be sequentially end-labeled.

DOI: https://doi.org/10.7554/eLife.44489.010

negative control, the same oligos were mixed together without the target strand. Fluorescence emission spectra were measured for the three samples and FRET efficiency was estimated from the peak donor and acceptor intensities (at 565 and 670 nm, respectively; see Materials and methods). The end-labeled Ndc80 complex exhibited a FRET efficiency of 0.29 (*Figure 2E*), a value intermediate between the positive and negative controls (0.53 and 0.10, respectively). This observation indicates that in solution, the Ndc80 complex can adopt a tightly bent conformation, which brings its two ends into close proximity.

## The Ndc80 complex fluctuates between tightly bent and more open conformations

Simple bulk fluorescence measurements cannot distinguish between a single population of Ndc80 complexes exhibiting moderate FRET or a mixture of complexes including both low- and high-FRET subpopulations, nor can they provide information about dynamics of the underlying conformational changes. We therefore turned to single molecule FRET to examine bending of individual complexes. End-labeled Ndc80 complexes (*Figure 3—figure supplement 1A and B*) were specifically tethered

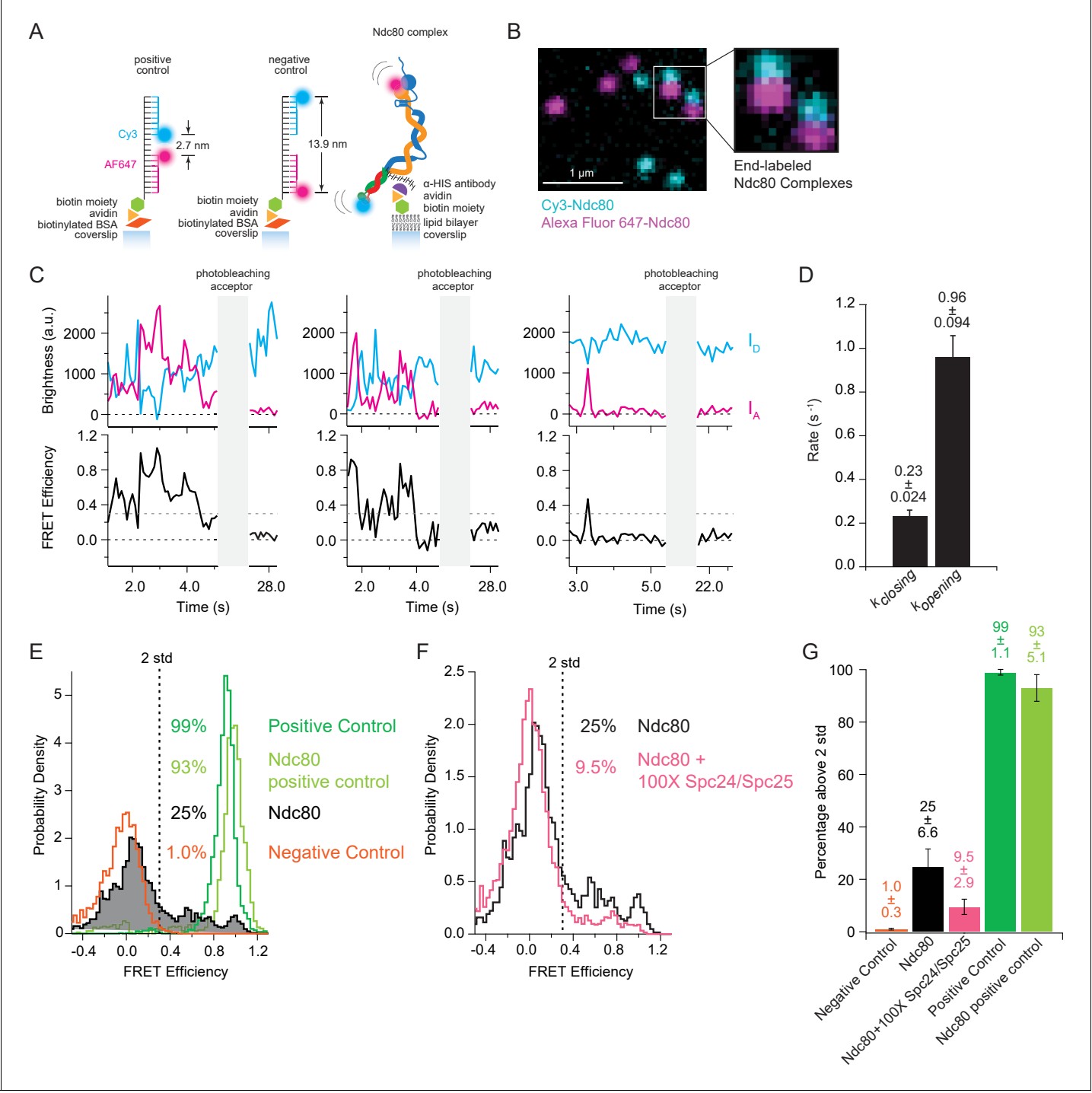

**Figure 3.** The Ndc80 complex fluctuates between tightly bent and more open conformations. (**A**) (*Left*) Cartoon depicting method of tethering labeled control oligonucleotides to glass coverslip. (*Left* and *middle*) Cartoons show the distance in nanometers between FRET pair dyes for positive and negative control oligonucleotides. (*Right*) Cartoon depicting method of tethering end-labeled Ndc80 complex to coverslip via lipid bilayer. (**B**) Fluorescence image showing coverslip decorated with end-labeled Ndc80 Nuf2 S2C Spc25 S154C complexes. Zoom-in shows two Ndc80 complexes with both dyes present. Colors are off-set vertically. (**C**) (*Top*) Fluorescence traces for three examples of end-labeled Ndc80 complexes (first two panels, Ndc80 Nuf2 S2C Spc24 N185C, last panel Ndc80 Nuf2 S2C Spc25 S154C) before and after acceptor (Alexa Fluor 647) photobleaching. (*Bottom*) Corresponding traces of FRET efficiency before and after photobleaching acceptor for each top example. Black dotted line indicates 0.0 FRET efficiency. Gray dotted line indicates threshold (0.30) used for two-state thresholding analysis. (**D**) Bar graph of switching rates calculated for Ndc80 complex Nuf2 S2C Spc25 S154C from a two-state thresholding analysis. Full data set analyzed is included in *Figure 3—source data 1*. (**E**) Histograms of FRET values for positive oligonucleotide control (n = 2699, 75 particles), Ndc80 Spc25 S154C G177C-positive control (n = 1275, 60 particles), Ndc80

*Figure 3 continued on next page*

*Figure 3 continued*

complex Nuf2 S2C Spc25 S154C (n = 2099, 85 particles) and negative oligonucleotide control (n = 3833, 154 particles). Percentage values represent the percentage of FRET values two standard deviations away from the mean of a Gaussian fit to the negative oligonucleotide data. 'n' refers to each 0.1 s FRET measurement for each condition. (F) Histograms of FRET values for Ndc80 complex Nuf2 S2C Spc25 S154C (n = 2099, 85 particles) (repeated from *Figure 3E*) and Ndc80 complex Nuf2 S2C Spc25 S154C + 100X Spc24/Spc25 (n = 4137, 152 particles). Percentage values represent the percentage of FRET values two standard deviations away from the mean of a Gaussian fit to the negative oligonucleotide data. 'n' refers to each 0.1 s FRET measurement for each condition. (G) Bar graph of the percentage of FRET values above two standard deviations away from the mean of a Gaussian fit to the negative oligonucleotide data. Error calculated as standard error of the mean of day to day variability of percentage above two standard deviations. The corrected FRET values included in each FRET efficiency histogram are included in *Figure 3—source data 2*. Additional supplementary data are included in *Figure 3—figure supplement 1*.

DOI: https://doi.org/10.7554/eLife.44489.012

The following source data and figure supplements are available for figure 3:

**Source data 1.** Analysis of threshold crossing from records of Ndc80 FRET versus time.

DOI: https://doi.org/10.7554/eLife.44489.015

**Source data 2.** All individual FRET values measured for oligonucleotide controls and Ndc80 complexes.

DOI: https://doi.org/10.7554/eLife.44489.016

**Figure supplement 1.** Concurrent labeling of the Ndc80 complex is used for single molecule FRET assays.

DOI: https://doi.org/10.7554/eLife.44489.013

**Figure supplement 1—source data 1.** Ratios of donor emission enhancement after photobleaching the acceptor dyes versus before photobleaching, measured for individual Ndc80 complexes and oligonucleotide control molecules.

DOI: https://doi.org/10.7554/eLife.44489.014

to coverslip surfaces and imaged by multi-color TIRF microscopy (*Figure 3A and B*). Emission from individual donor (Cy3) and acceptor (Alexa Fluor 647) dyes was measured simultaneously from brief, ~2 s time-lapse recordings during excitation of the donor (i.e. at a wavelength of 561 nm). FRET versus time for each complex was computed from the donor and acceptor signals, $I_D$ and $I_A$, after subtracting background levels and correcting for spillover and cross-excitation (See Materials and methods) (*Roy et al., 2008*; *Selvin and Ha, 2008*) (*Figure 3C*). Following each measurement, the acceptor was excited directly (at 641 nm) until it photobleached, to confirm the presence of a single acceptor dye on every analyzed complex (*Figure 3—figure supplement 1C*).

Donor and acceptor signals recorded from individual end-labeled Ndc80 complexes were anti-correlated; their FRET levels fluctuated between low values around zero and higher values approaching unity (*Figure 3C*). The distribution of FRET levels was broad, with a large fraction (25%) of high-FRET values more than two standard deviations above the mean of the no-FRET population (*Figure 3E*). In contrast, FRET from positive and negative control oligos was more stable, with only small deviations from unity and zero, respectively (*Figure 3E* and *Figure 3—figure supplement 1D and F*). Likewise, FRET from a positive control Ndc80 complex, which carried donor and acceptor dyes close together on the same face of Spc25 (*Figure 3—figure supplement 1E*), remained stably high (*Figure 3—figure supplement 1F*). Based on a simple two-state threshold-crossing analysis, switching of the end-labeled Ndc80 complexes from low to high FRET occurred at a rate of $k_{closing}$ = 0.23 ± 0.024 s$^{-1}$. Switching from high to low FRET was about fourfold faster, occurring at a rate of $k_{opening}$ = 0.96 ± 0.094 s$^{-1}$ (*Figure 3D*). The high-FRET fluctuations were eliminated when intra-complex interactions were competitively inhibited by adding a hundredfold excess of free Spc24/Spc25 (*Figure 3F and G*). Altogether, these single molecule observations confirm that the Ndc80 complex can adopt a tightly bent conformation, with both ends interacting closely together, in agreement with our bulk measurements, and they show that the complex fluctuates dynamically between tightly bent and more open conformations.

## The loop region of Ndc80 contributes to tight bending of the complex

A flexible region of 50–60 residues within the Ndc80 protein interrupts the coiled-coil of Ndc80/Nuf2 to form a loop (*Ciferri et al., 2005*), which has been implicated in flexibility of the Ndc80 complex (*Wang et al., 2008*) and might serve as a binding site for other proteins (*Maure et al., 2011*). To determine whether this region also contributes to tight bending, we measured FRET levels for a mutant Ndc80 Δ490–510 complex lacking part of the loop (*Figure 4A and B*). In budding yeast cells, this partial loop deletion confers temperature sensitivity (*Maure et al., 2011*), indicating functional

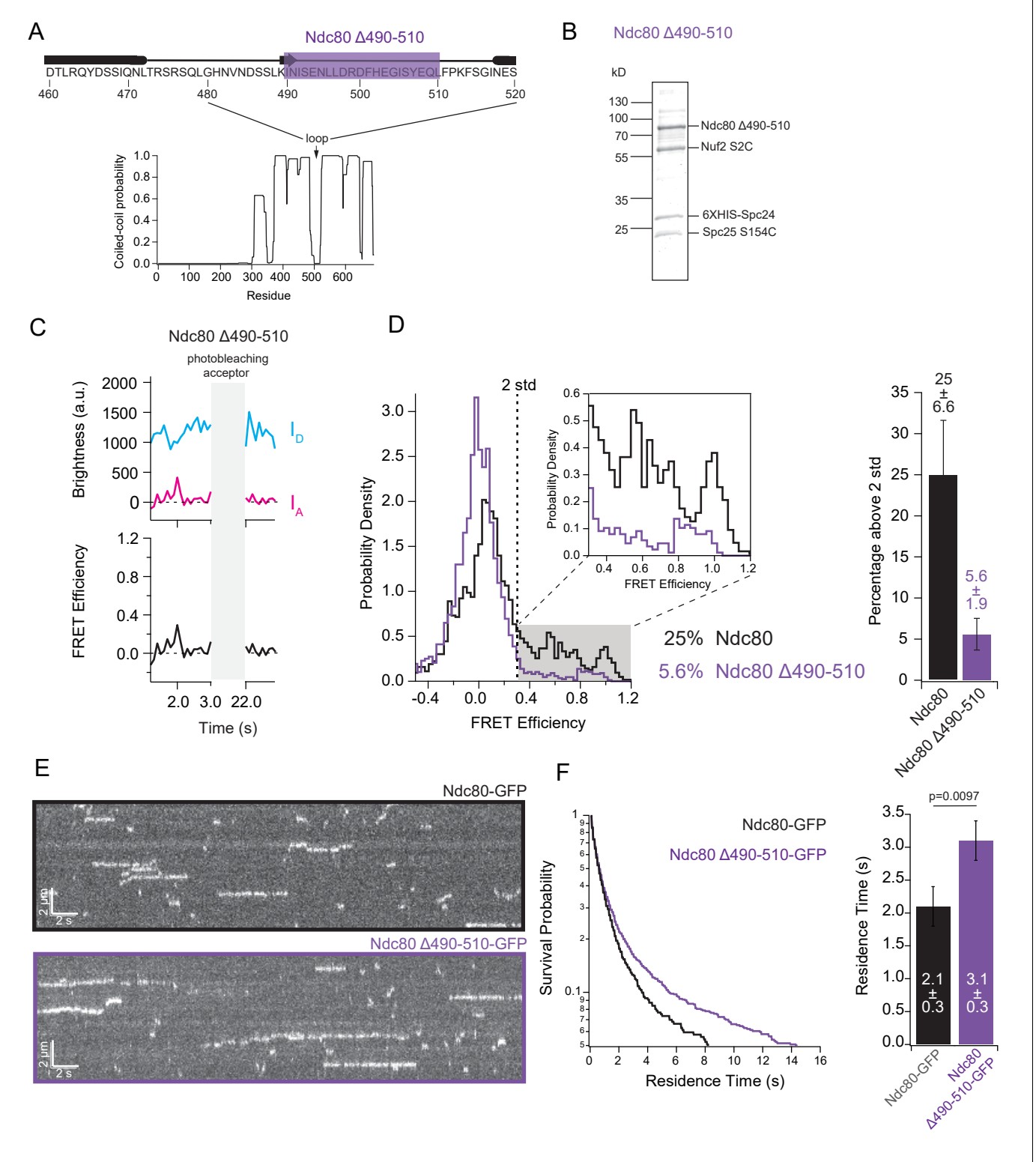

**Figure 4.** The loop region of Ndc80 contributes to tight bending of the complex. (**A**) (*Top*) Primary sequence of Ndc80 protein amino acids 460–520. The purple bar highlights the region of the loop deleted for the following experiment. (*Bottom*) Graph of probability of coiled-coil as predicted by PCOILS of the Ndc80 protein. (**B**) Coomassie blue-stained gel showing end-labeled Ndc80 Δ490–510 complex used in single molecule FRET assay. (**C**) (*Top*) Fluorescence traces of one representative example of an end-labeled Ndc80 complex in *Figure 4C* before and after photobleaching acceptor (Alexa Fluor 647). (*Bottom*) Corresponding traces of FRET efficiency before and after acceptor photobleaching for the top example. (**D**) (*Left*)

*Figure 4 continued*

Histograms of FRET values for Ndc80 complex Nuf2 S2C Spc25 S154C (n = 2099, 85 particles) (data repeated from *Figure 3E and F*) and Ndc80 complex Ndc80 Δ490–510 Nuf2 S2C Spc25 S154C (n = 2926,143 particles). Percentage values represent the percentage of FRET values two standard deviations away from the mean of a Gaussian fit to the negative oligonucleotide data. 'n' refers to each 0.1 s FRET measurement for each condition. Zoom in of histograms depicting FRET values over two standard deviations. (*Right*) Bar graph of the percentage of FRET values above two standard deviations. Error calculated as standard error of the mean of day-to-day variability of percentage above two standard deviations. The corrected FRET values included in each FRET efficiency histogram are included in *Figure 4—source data 1*. (E) Representative kymographs of Ndc80-GFP and Ndc80 Δ490–510-GFP. Brightness and contrast have been adjusted separately for each image for best visualization. (F) (*Left*) Survival probability curves of residence times for Ndc80-GFP (n = 570) (black trace) and Ndc80 Δ490–510 (n = 1237) (purple trace). (*Right*) Bar graph of average residence times of the data at left. Error was calculated using bootstrapping analysis. p-Value was calculated using a two-tailed Student's *t* test. Raw data of all residence times are included in *Figure 4—source data 1*. Additional supplementary data are included in *Figure 4—figure supplement 2*.
DOI: https://doi.org/10.7554/eLife.44489.017

The following source data and figure supplements are available for figure 4:

**Source data 1.** All individual FRET values and residence times on microtubules measured for individual mutant Ndc80 Δ490-510 loop-deletion complexes.
DOI: https://doi.org/10.7554/eLife.44489.021
**Figure supplement 1.** Model of the Ndc80 complex with all combinations of dye locations.
DOI: https://doi.org/10.7554/eLife.44489.018
**Figure supplement 2.** The Ndc80 complex has preferred orientations for intra-complex interactions.
DOI: https://doi.org/10.7554/eLife.44489.019
**Figure supplement 2—source data 1.** All individual FRET values measured for individual Ndc80 complexes with four pairwise combinations of dye locations.
DOI: https://doi.org/10.7554/eLife.44489.020

importance. FRET levels from purified individual Ndc80 Δ490–510 complexes were relatively low. The percentage of high-FRET values was only 5.6%, which is fourfold less than the percentage measured from wild-type complexes (*Figure 4C and D*), indicative of a more open form of the Ndc80 complex. According to our model, a more open conformation of the complex should bind microtubules better. Indeed, Ndc80 Δ490–510 exhibited a longer residence time on microtubules than full-length wild type complex (3.1 ± 0.3 s vs. 2.1 ± 0.3 s; *Figure 4E and F*). These observations confirm that the loop region contributes to tight bending of the Ndc80 complex and suggest that flexibility of the complex directly influences its microtubule binding.

We also examined which areas on the globular ends of the Ndc80 complex come close together in the tightly bent conformation by creating an additional three end-labeled Ndc80 complexes. We chose two new sites within the complex to mutate to cysteine residues (Nuf2 Q74C and Spc24 N185C), and along with the two original dye locations (Nuf2 S2C and Spc25 S154C), engineered pairwise combinations, each with one dye on the Ndc80/Nuf2 end and one dye on the Spc24/Spc25 end (*Figure 4—figure supplement 1*). Retaining one dye at the original location on Nuf2 S2C and moving the other to Spc24 N185C did not significantly change the FRET levels (25 ± 6.6% high-FRET values versus 25 ± 5.2% for the original locations) (*Figure 4—figure supplement 2A and B*). However, moving the dye from Nuf2 S2C to Nuf2 Q74C, regardless of the dye location on Spc24/Spc25, reduced the fraction of high-FRET values more than twofold (*Figure 4—figure supplement 2B and C*). These results suggest that, in the tightly bent configuration, the Spc24/Spc25 end of the complex associates closely with one face of the Ndc80/Nuf2 end, near the N-terminus of Nuf2.

## Tight bending of the Ndc80 complex is inhibited by binding to either microtubules or the MIND complex

If tight bending of the Ndc80 complex inhibits microtubule binding, then Ndc80 complexes bound to microtubules should be found primarily in the more open conformation. (I.e. binding to microtubules should 'select' for the open complexes.) To test this prediction, we measured FRET levels for individual end-labeled complexes bound to taxol-stabilized microtubules (*Figure 5A*). As expected, FRET from the microtubule-bound complexes was low, with only a small fraction of high-FRET values, 6.2% (*Figure 5B and D*). Assuming a simple four-state equilibrium between closed and open configurations (*Figure 5E*), the high-FRET fractions measured for microtubule-attached (*Figure 5D*) and 'free' Ndc80 complexes (coverslip-tethered, *Figures 3D*, *4C* and *5D*) imply that the tightly bent

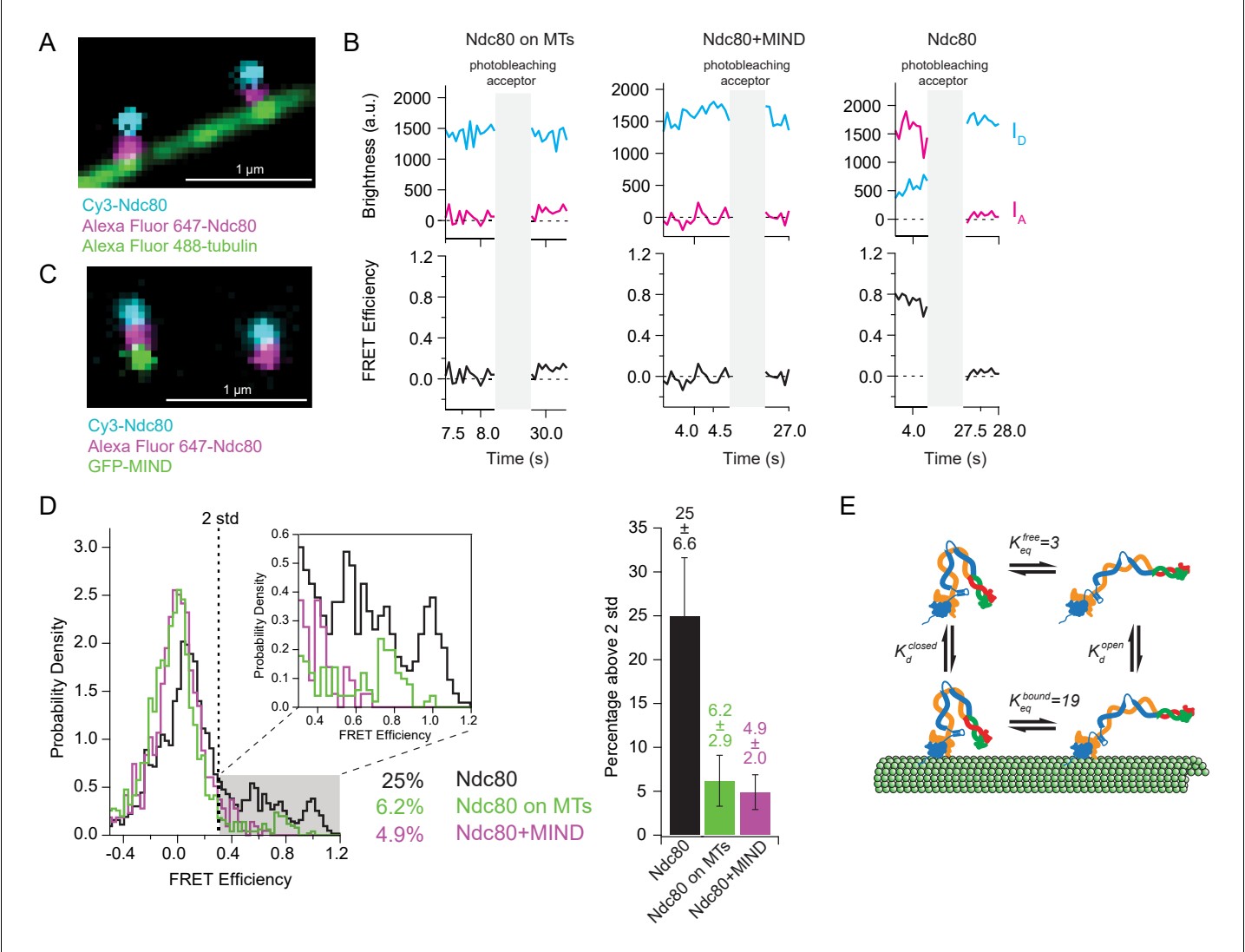

**Figure 5.** Tight bending of the Ndc80 complex is inhibited by binding to either microtubules or the MIND complex. (A) Fluorescence image of two end-labeled Ndc80 complexes (Nuf2 S2C Spc25 S154C) on Alexa Fluor 488 labeled-microtubules. Colors are off-set vertically. (B) (*Top*) Fluorescence traces of one representative example of an end-labeled Ndc80 complex in each condition shown in *Figure 5D* before and after photobleaching acceptor (Alexa Fluor 647). (*Bottom*) Corresponding traces of FRET efficiency before and after acceptor photobleaching for each top example. (C) Fluorescence image of two end-labeled Ndc80 complexes (Nuf2 S2C Spc25 S154C) next to each other on one coverslip, colocalized with MIND-GFP on the left, but not on the right. Colors are off-set vertically. (D) (*Left*) Histograms of FRET values for Ndc80 complex Nuf2 S2C Spc25 S154C (n = 2099, 85 particles) (data repeated from *Figure 3E*, *Figure 4C*), Ndc80 complex Nuf2 S2C Spc25 S154C on microtubules (n = 1682, 91 particles) and Ndc80 complex Nuf2 S2C Spc25 S154C bound to MIND (n = 718, 41 particles). Percentage values represent the percentage of FRET values two standard deviations away from the mean of a Gaussian fit to the negative oligonucleotide data. 'n' refers to each 0.1 s FRET measurement for each condition. Zoom in of histograms depicting FRET values over two standard deviations. (*Right*) Bar graph of the percentage of FRET values above two standard deviations. Error calculated as standard error of the mean of day-to-day variability of percentage above two standard deviations. The corrected FRET values included in each FRET efficiency histogram are included in *Figure 5—source data 1*. (E) Diagram of equilibrium between open and closed conformations of the Ndc80 complex, on and off the microtubule. Estimation of $K_{eq}$ values described in Materials and methods. Additional supplementary data are included in *Figure 5—figure supplement 1*.

DOI: https://doi.org/10.7554/eLife.44489.022

The following source data and figure supplements are available for figure 5:

**Source data 1.** All individual FRET values measured for individual Ndc80 complexes bound to microtubules or bound to the MIND complex.
DOI: https://doi.org/10.7554/eLife.44489.025

**Figure supplement 1.** Reproducibility of Ndc80 complex single molecule FRET measurements.
DOI: https://doi.org/10.7554/eLife.44489.023

*Figure 5 continued on next page*

*Figure 5 continued*

**Figure supplement 1—source data 1.** All individual FRET values measured for individual Ndc80 complexes during technical and biological replicate experiments.

DOI: https://doi.org/10.7554/eLife.44489.024

conformation has a sixfold lower affinity for microtubules than the open conformation (see Materials and methods).

Association of the Ndc80 and MIND complexes together creates a co-complex with enhanced affinity for microtubules relative to Ndc80 complex alone (*Kudalkar et al., 2015*). Genetic evidence suggests this enhancement might occur because MIND prevents the Ndc80 complex from adopting the tightly bent, auto-inhibited state (*Tien et al., 2014*; *Kudalkar et al., 2015*). The Ndc80 complex of *C. elegans* might also be activated by the Mis12 complex, which is the worm counterpart of MIND (*Cheeseman et al., 2006*). To test directly whether MIND prevents tight bending, we added free GFP-MIND to the end-labeled Ndc80 complexes (which were tethered sparsely to coverslips, as above). Using three-color TIRF microscopy, the end-labeled Ndc80 complexes that formed co-complexes with MIND were first distinguished from those that did not by assessing co-localization of the GFP signal with both Cy3 and Alexa Fluor 647 dyes (*Figure 5C*). Then FRET between donor and acceptor dyes on the Ndc80 complexes was measured. With a tenfold excess of free GFP-MIND, 33% of the end-labeled Ndc80 complexes formed co-complexes with MIND. FRET levels from these Ndc80-MIND co-complexes were low, with only 4.9% of values significantly above the no-FRET population (*Figure 5B and D*). FRET levels from Ndc80 complexes in the same fields of view but lacking MIND-GFP were higher, with a fivefold larger fraction of high-FRET values, 29%, which is statistically indistinguishable from the high-FRET fraction measured for Ndc80 complexes alone, in the absence of GFP-MIND (*Figure 5—figure supplement 1A and B*). These observations show that association of the Ndc80 complex with MIND inhibits tight bending of the Ndc80 complex.

## Discussion

In this study, we have directly examined the functional significance of conformational changes of the Ndc80 complex in vitro. Using single molecule techniques, we found that the isolated Ndc80 complex fluctuates dynamically around its hinge region, adopting both open and closed conformations. Tight bending leads to an auto-inhibited state that decreases microtubule affinity. MIND-binding relieves this auto-inhibition, which explains how MIND enhances binding of the Ndc80 complex to microtubules. Our data reveal a previously uncharacterized mechanism of regulation of the Ndc80 complex that is distinct from the more well-known phospho-regulation of the complex.

Our findings also provide an explanation for prior observations made in vivo. Deleting part of the loop region from yeast Ndc80 causes temperature-sensitive cell growth and frequent failures in the biorientation of sister kinetochores (*Maure et al., 2011*). Likewise, a pair of point mutations near the loop region of Ndc80 causes temperature-sensitive growth and defects in biorientation (*Tien et al., 2014*). These defects in vivo were suppressed by additional cis-acting mutations, in either Ndc80 or Nuf2, made on exactly the opposite side of the loop region, suggesting that they interfere with tight bending of the Ndc80 complex (*Tien et al., 2014*). Moreover, the same loop-proximal point mutations enhance microtubule-binding of the Ndc80 complex in vitro, in a manner that mimics the addition of MIND (*Kudalkar et al., 2015*). Based on our new findings, all these prior observations could have a single mechanistic basis: Tight bending of the Ndc80 complex might be required for proper regulation of kinetochore-microtubule attachments in vivo.

We established that MIND opposes the tightly bent form of the Ndc80 complex, but precisely how this translates to increased microtubule binding is less clear, given that MIND interacts with a region far away from the microtubule-binding domains of the Ndc80 complex. One possibility is that the mechanism is similar to the auto-inhibition of kinesin-1: kinesin-1 bends around its loop to bring its tail in close proximity to its heads, thereby inhibiting motor activity of the heads (*Kaan et al., 2011*). Binding of the kinesin-1 tail to cargo blocks its interaction with the heads, thereby relieving auto-inhibition and enabling processive motility. Like the cargo of kinesin-1, MIND binding to one end of the Ndc80 complex could simply block its interaction with the other end. However, our data suggest that MIND may be doing more than simply relieving auto-inhibition: while the Broccoli

construct (which lacks the Spc24/Spc25 dimer and is not auto-inhibited) resides on the microtubule lattice longer than the full-length Ndc80 complex, the Ndc80-MIND co-complex has a residence time still roughly twofold higher than Broccoli. This additional enhancement suggests that MIND might propagate a specific molecular reorganization to the Ndc80/Nuf2 end, through the coiled coil, to establish some preferred arrangement for microtubule binding. Allosteric communication through a coiled-coil has been seen, for example, in dynein (*Carter et al., 2008*).

Previous literature determined that the Ndc80 complex is flexible (*Wang et al., 2008*) due to its internal loop region (*Zhang et al., 2012*) and can exist in both a bent and more open state (*Tien et al., 2014*). Our data now show that it can readily adopt an extremely bent state, in which the two ends of the complex come within a few nanometers of each other, and that the loop region is required for this tight bending. The Ndc80 complex is commonly drawn as a rigid rod containing a single, distinct hinge point that coincides with the loop region, which is located roughly 20 nm from the Ndc80/Nuf2 end and 40 nm from the Spc24/Spc25 end of the complex (*Wang et al., 2008*). Given these distances, however, a simple hinging about the loop region would not bring the two ends close enough together to explain the high FRET values we observed (*Wang et al., 2008*). Thus, additional sites of flexibility must exist within the complex. Flexibility at the junction between the calponin homology domains and the hairpin/coiled coil region has been suggested based on the different configurations of this region in crystal structures (*Valverde et al., 2016*; *Ciferri et al., 2008*). Additional hinging at this second region might be sufficient to bring the two ends close enough to explain the high FRET.

Tight bending of the Ndc80 complex is important for its function in vivo (*Tien et al., 2014*) and our observations now show that this conformation is auto-inhibited. However, the specific physiological role of auto-inhibition remains uncertain. One attractive possibility is that it might prevent newly expressed Ndc80 complexes from interacting with microtubules prematurely, before the Ndc80 complexes are anchored at a kinetochore, thereby avoiding delay or potentially persistent mis-localization. Another, not mutually exclusive idea is that auto-inhibition might ensure stepwise assembly of the outer kinetochore. Kinetochores are thought to assemble hierarchically, with the inner-most components assembling first and then outer components building sequentially upon them, according to their spatial arrangement (*Musacchio and Desai, 2017*). Such stepwise assembly would not occur if all components were continuously able to bind their partners with high affinity. However, it could be explained by a sequential relief of auto-inhibition as each component assembles onto a nascent kinetochore. Previous work shows that binding of MIND to its kinetochore receptors, Ame1 and Mif2, is prevented by auto-inhibition, presumably until MIND binds a nascent kinetochore, where its auto-inhibition can be relieved by Aurora phosphorylation. Similarly, the auto-inhibition of Ndc80 complex might prevent it from recruiting the Dam1 complex until it is bound to kinetochore-anchored MIND. Sequential relief of auto-inhibition might be a general mechanism that encodes stepwise construction at various sites within the kinetochore (*Dimitrova et al., 2016*; *Petrovic et al., 2016*).

Some Ndc80 complex is localized to kinetochores independently of the MIND complex and anchored instead via Cnn1/CENP-T (*Malvezzi et al., 2013*). Determining whether or not Cnn1 binding promotes opening of the Ndc80 complex and relieves its auto-inhibition could be an interesting area for future work. If some kinetochore-anchored Ndc80 complexes can adopt the tightly bent state, then their conformation might be sensitive to kinetochore tension, potentially contributing to the mechanosensory feedback mechanisms that are thought to control error correction and checkpoint signaling at kinetochores.

## Materials and methods

### Key resources table

| Reagent type or resource | Designation | Source or reference | Identifiers | Additional information |
|---|---|---|---|---|
| Strain, *E. coli* | Rosetta (DE3) competent cells | Millipore Sigma | Cat#70954 | |

*Continued on next page*

*Continued*

| Reagent type or resource | Designation | Source or reference | Identifiers | Additional information |
|---|---|---|---|---|
| Biological sample | Bovine brain tubulin | lab purification | | Protocol adapted from *Castoldi and Popov, 2003*. |
| Sequence-based reagent | Fluorescently-labeled oligonucleotides | IDT | | |
| Sequence-based reagent | Fluorescently-labeled, biotintylated oligonucleotides | IDT | | |
| Antibody | Penta-HIS biotin conjugate, monoclonal mouse | Qiagen | Cat#34440 | Diluted to 20 nM |
| Chemical compound | Alexa Fluor 488 succinimidyl ester | Thermo Fisher | Cat#A20000 | |
| Chemical compound | Alexa Fluor 647 maleimide | Thermo Fisher | Cat#A20347 | |
| Chemical compound | Alexa Fluor 568 succinimidyl ester | Thermo Fisher | Cat#A20003 | |
| Chemical compound | Cy3 maleimide | GE Healthcare Life Sciences | Cat#PA23031 | |
| Chemical compound | Cy5 maleimide | GE Healthcare Life Sciences | Cat#PA25031 | |
| Chemical compound | Trolox | Millipore Sigma | Cat#238813 | |
| Chemical compound | Glucose oxidase | Millipore Sigma | Cat#345386 | |
| Chemical compound | Catalase | Millipore Sigma | Cat#219261 | |
| Chemical compound | POPC | Avanti Polar Lipids | Cat#850457 | 1-palmitoyl-2-oleoyl-sn-glycero-3-phosphocholine |
| Chemical compound | Bio-cap-PE/BioPE | Avanti Polar Lipids | Cat#870273 | 1,2-dioleoyl-sn-glycero-3-phosphoethanolamine-N-(cap biotinyl) sodium salt |
| Chemical compound | Biotinylated bovine serum albumin (BSA) | Vector Laboratories | Cat#B-2007 | |
| Chemical compound | Avidin DN | Vector Laboratories | Cat#A-3100 | |
| Chemical compound | TCEP | Thermo Fisher | Cat#T2556 | |
| Chemical compound, drug | Paclitaxel/Taxol | Millipore Sigma | Cat#T7402 | |
| Software | Labview | National Instruments | | |
| Software | Igor Pro | Wavemetrics | | |

## Protein expression and purification

All protein complexes were derived from *S. cerevisiae* but expressed recombinantly in *E. coli* and purified in essentially two steps, beginning with His6- or FLAG-based immuno-precipitation, followed by size exclusion chromatography. The full, heterotetrameric Ndc80 complexes each carried a single His6 tag on either the N- or C-terminus of Spc24 (i.e. Spc24-His6-tagged or His6-Spc24-tagged complex). Likewise, the hetero*dimeric sub*complexes (i.e., His6-Spc24/Spc25 and the 'Broccoli' construct, His6-Ndc80/Nuf2-GFP) each carried an N-terminal His6 tag on one subunit. Each heterodimeric subcomplex was encoded on a separate, dicistronic vector (petDuet and pRSF, for Ndc80/Nuf2 and Spc24/Spc25, respectively), both of which were co-transfected into BL21 cells for purification of the full, heterotetrameric complexes. For purification of subcomplexes alone, single dicistronic vectors were transfected individually. Expression was induced with 0.2 mM isopropyl β-D-

1-thiogalactopyranoside (IPTG) for 14 hr at 20°C, cells were lysed with a French press in buffer H (50 mM HEPES, 200 mM NaCl, pH 7.6) supplemented with protease inhibitors (Roche; Basel, Switzerland), 5 mM imidazole, 0.5 uL benzonase and 1 mM PMSF. The cell lysate was then immuno-precipitated onto a 5 mL Ni-charged IMAC resin column (Bio-Rad; Hercules, CA), washed with buffer H, and eluted with 400 mM imidazole in buffer H. The eluate was loaded onto a Superdex 200 16/60 column (GE Healthcare; Chicago, IL) for size-exclusion chromatography. Fractions containing the complex were identified by UV-absorption and Coomassie blue-stained SDS-PAGE, pooled, and protein concentration was measured using bicinchoninic acid (BCA) (Sigma; St. Louis, MO). We note that three attempts to purify a full-length, heterotetrameric Ndc80 complex carrying the charge-reversal 7K mutations in the Ndc80 protein failed to produce a stable heterotetramer, suggesting that the 7K mutant might destabilize the complex. In contrast, the dimeric 7K Spc24/Spc25 was biochemically well-behaved.

MIND complex carrying an N-terminal FLAG tag on Nsl1 and a C-terminal GFP on Mtw1 was encoded on a single, IPTG-inducible polycistronic vector. Transfected BL21 cells were induced and lysed using a French press, essentially as described above, with the following changes: lysis, immuno-precipitation, and chromatography were all carried out in buffer N (50 mM NaPO$_4$, 200 mM NaCl, pH 7.0). Immuno-precipitation used an anti-FLAG M2 affinity gel (A2220; Sigma; St. Louis, MO). Protein was eluted from the anti-FLAG gel using 0.1 mg/ml 3X FLAG Peptide (F4799; Sigma; St. Louis, MO) in buffer N, and then size exclusion chromatography was performed as described above, but in buffer N.

MIND-GFP/Ndc80 co-complex was prepared by combining two nickel-purified complexes (MIND-GFP and Ndc80) and then performing size-exclusion chromatography. In this case, the MIND complex carried a C-terminal His6 tag on Dsn1 and a C-terminal GFP tag on Mtw1. Expression, lysis, and immuno-purification were performed as described above, using buffer H for Ndc80 and buffer N for MIND. Then the nickel-purified complexes were mixed in a 2.5:1 molar ratio (MIND:Ndc80), incubated for 15 min at room temperature, and purified with a Sepharose 400 size exclusion column using buffer S (50 mM NaPO$_4$, 100 mM NaCl, pH 7.0) (GE Healthcare; Chicago, IL).

## Protein labeling for FRET studies

For bulk FRET measurements, the two heterodimeric subcomplexes of the Ndc80 complex were expressed separately, using the methods described above, from dicistronic vectors that were mutated to remove all eight native cysteines and to add the new cysteines required for site-directed labeling with thiol-reactive dyes. After the 'cysteine-light' His6-Spc24/Spc25 subcomplex was bound to nickel resin and washed, it was incubated overnight with ~30 mM Cy3 maleimide (GE Healthcare; Chicago, IL). The resin was then washed again, to remove free Cy3, and added to cleared lysate from cells expressing the (cysteine-light) Ndc80/Nuf2 subcomplex, to allow formation of the full heterotetrameric Ndc80 complex. The resin was then washed and incubated overnight with ~30 mM Cy5 maleimide (GE Healthcare; Chicago, IL). Proteins were eluted with 400 mM imidazole in buffer H and then purified by size exclusion chromatography (using the Superdex 200 column, as described above). Labeling efficiency was estimated to be 1:1 Cy3:Spc45/Spc25 dimer and 0.8:1 Cy5:Ndc80/Nuf2 dimer by calculating protein concentration (using a NanoDrop and BCA assay) and calculating dye molecule concentration (using a NanoDrop).

For single molecule FRET assays, the two cysteine-light heterodimeric subcomplexes were co-expressed together, according to the methods described earlier. After the full heterotetrameric Ndc80 complex was bound to the nickel resin and washed, it was incubated overnight with an equimolar combination of ~15 mM Cy3 maleimide and ~15 mM Alexa Fluor 647 maleimide (Thermo Fisher; Waltham, MA). The complex was eluted with 400 mM imidazole in buffer H and then purified by size exclusion chromatography (Superdex 200). Labeling with both dyes simultaneously during purification decreased the duration of the protocol relative to the bulk FRET protocol (where dyes were added sequentially), thereby avoiding degradation of the N-terminal Ndc80 tail (*Figure 3—figure supplement 1A and B*). Our labeling protocol was adapted from *Joo and Ha (2012)*.

## Bulk FRET measurement

Dye-labeled oligonucleotides were purchased (from IDT; Coralville, IA) with the sequences 5'-GCTA TGACCATGATATAC-Cy5-3' and 5'-Cy3-AGCGCGCAATTAACCC-3' (*Tsuji et al., 2001*). To create a

positive control, the dye-labeled oligos were annealed in buffer T (10 mM TRIS buffer pH 7.5, 50 mM NaCl) onto a complementary unlabeled target strand, 5'-GGGTTAATTGCGCGCTTGGCGTAA TCATGGTCATAGC-3', by mixing 10 uM of the target strand with 5 uM each of the single-stranded, labeled oligonucleotides, heating at 95°C for 2 min, and then allowing the mixture to cool to room temperature for 1 hr. For the negative control, the two labeled oligos were mixed together without the target strand. To determine bulk FRET levels, oligo controls or end-labeled Ndc80 complexes were diluted to 0.25 uM in buffer H and fluorescence emission spectra were measured from 560 to 730 nm in a spectrofluorometer (Spex Fluorolog-3, Horiba Jobin Yvon; Edison, NJ), under 550 nm excitation and using a 2 nm bandwidth. Relative FRET efficiency, $E_{FRET}$, was calculated using the equation,

$$E_{FRET} = \frac{I_A}{I_A + I_D}$$

where $I_D$ and $I_A$ represent the background-subtracted intensities of donor (Cy3) and acceptor (Cy5) dyes, measured at their peak emission wavelengths (565 and 670 nm, respectively).

## Preparation of oligo controls and flow channel setup for single molecule FRET

For tethering onto streptavidin-coated coverslip surfaces, a biotinylated oligonucleotide target strand was purchased with the sequence, 5'-Biotin-GGGTTAATTGCGCGCTTGGCAAAAGTATCA TGGTCATAGC-3'. The positive control was created in this case with two additional dye-labeled oligos, purchased with the sequences 5'-GCTATGACCATGATATAC-AF647-3' and 5'-Cy3-AGCGCG-CAATTAACCC-3'. When both of these are annealed to the target strand, the donor and acceptor dyes (Cy3 and Alexa Fluor 647, respectively) are held in very close proximity (<10 nm). For the negative control, the same two oligos were ordered but with the dyes moved to opposite ends, 5'-AF 647-GCTATGACCATGATATAC-3' and 5'-AGCGCGCAATTAACCC-Cy3-3'. When both of these are annealed to the target strand, the two dyes are held far apart (>10 nm). Annealing was performed in buffer T using the same method described above for bulk FRET. Alexa Fluor 647 was used as the acceptor dye for all single molecule FRET measurements (i.e. on oligonucleotide controls as well as on the Ndc80 complexes) because of its greater photostability compared to Cy5.

Control, surface-tethered oligos were observed in flow channels created using double-stick tape adhered to plasma-cleaned glass slides and coverslips (Deng and Asbury, 2017). Briefly, dry channels were assembled and then warmed for at least 3 hr at 50 °C, to promote adherence of the tape to the glass. Biotinylated bovine serum albumin (Vector Laboratories; Burlingame, CA) was introduced first and incubated for 5 min. After a wash in buffer TE (100 mM Tris, pH 8, 10 mM EDTA, 100 mM KCl), avidin-DN (Vector Laboratories; Burlingame, CA) was introduced and incubated for 5 min. The channel was re-washed with buffer TE supplemented with 8 mg/ml BSA, and then the oligos were introduced and incubated for 3 min, to allow binding to the surface. Untethered oligos were removed by a final wash with buffer TE supplemented with 8 mg/ml BSA plus an oxygen scavenging system (2 μM glucose oxidase, 10 μM catalase, 10 mM Trolox, 1 mM glucose). The channels were sealed with nail polish and imaged in a TIRF microscope, as described below (under single molecule FRET microscopy).

Surface-tethered Ndc80 complexes were observed in flow channels created as described above but using PEGylated coverslips. After warming at 50°C for 3 hr, the dry channels were first filled with buffer HN (25 mM HEPES buffer pH 7.6, 150 mM NaCl) and then small unilamellar vesicles (0.1% BioPE lipids in POPC lipids, Avanti Polar Lipids, Inc; Alabaster, AL) were introduced and incubated for 5 min, to coat the glass surfaces with a supported lipid bilayer that included a small fraction (0.1%) of biotinylated lipids (Deng and Asbury, 2017). We found that passivation with a supported lipid bilayer was necessary to prevent non-specific adsorption of end-labeled Ndc80 complexes onto the coverslip (whereas passivation with bovine serum albumin was sufficient for tethering the oligo controls, as described above). After washing with buffer HN, avidin-DN was introduced and incubated for 5 min. The channel was then washed with BRB80 (80 mM piperazine-N,N'-bis(2- ethanesulfonic acid), pH 6.9, 1 mM MgCl₂, 1 mM EGTA) supplemented with blocking proteins, 6.9 mg/ml BSA and 0.04 mg/ml κ-casein. Anti-penta-His antibody (Qiagen; Venlo, Netherlands) was then introduced and incubated for 5 min, and the channel was washed again with BRB80 plus blocking proteins. The

end-labeled Ndc80 complexes, diluted into BRB80 plus blocking proteins, were then introduced and incubated for 5 min to allow binding to the surface. Untethered Ndc80 complex was removed by a final wash of BRB80 plus blocking proteins and an oxygen scavenging system (2 µM glucose oxidase, 10 µM catalase, 1 mM glucose). The channels were sealed and imaged in a TIRF microscope as described below (under Single molecule FRET TIRF microscopy).

To observe surface-tethered Ndc80 complexes in co-complex with GFP-MIND, the end-labeled Ndc80 complex and GFP-MIND complex were pre-incubated in a 1:10 ratio for 15 min at room temperature, and then tethered to lipid-coated coverslips exactly as described in the previous paragraph.

To observe FRET from individual Ndc80 complexes on microtubules, Alexa Fluor 488-labeled, taxol-stabilized microtubules were tethered to coverslips, as previously described (*Powers et al., 2009*). The end-labeled Ndc80 complex was diluted in BRB60 buffer (60 mM piperazine-N,N'-bis(2-ethanesulfonic acid), pH 6.9, 1 mM MgCl$_2$, 1 mM EGTA), introduced and allowed to bind the tethered microtubules for 1 min before imaging. We used BRB60 rather than BRB80 for this experiment, because the reduction in ionic strength increases the residence time and slows the diffusion of the Ndc80 complexes on the microtubules (*Powers et al., 2009*), thereby facilitating collection of longer-duration records of fluorescence intensity from individual complexes.

## Single molecule FRET microscopy

All samples were imaged in a custom TIRF microscope, with three cameras that simultaneously recorded images in green (500 to 550 nm wavelengths), yellow (575 to 625 nm), and red (660 to 740 nm) color-bands, as described in detail in *Deng and Asbury (2017)*. In the following description of methods, we refer to each camera by the central wavelength of its color-band: 525 nm for green, 600 nm for yellow and 700 nm for red. Each prepared slide had up to three flow channels with different experimental conditions. One of the flow channels, containing all assembly reagents but lacking labeled oligonucleotides or protein, was always included as a check for background fluorescence. Movies lasting 30 or 40 s were recorded at 10 frames per second, with roughly 50 to 150 particles imaged per field of view, while the sample was illuminated in four distinct intervals: (i) first FRET was measured by exciting the donor (Cy3) and acceptor dyes (AF647) with 561 nm laser illumination (at a nominal power of 25 mW), (ii) then the acceptor dyes (AF647) were directly excited with 641 nm illumination (at 25 mW), (iii) then the 641 nm illumination power was increased (to 100 mW) in order to photobleach the acceptor dyes, and (iv) finally the donor dyes were measured alone with (25 mW) 561 nm illumination. This illumination sequence, which is shown in *Figure 3—figure supplement 1C*, allowed verification that both donor and acceptor dyes were present on every molecule analyzed, that donor fluorescence was enhanced after acceptor photobleaching, and that the correction of acceptor signals for cross-excitation and spillover was accurate. (The method of correcting for cross-excitation and spillover is described below.) For some experiments, an additional brief 488 nm laser illumination was added at the beginning and end of the sequence to record fluorescence from GFP-MIND or from Alexa Fluor 488-labeled microtubules.

## Single molecule FRET analysis

Analysis was done using custom, semi-automated software written in Labview (National Instruments; Austin, TX) and Igor Pro (Wavemetrics; Lake Oswego, OR). Source code is available upon request. For each movie (corresponding to a single field of view), the first 10 frames from each camera were averaged to create initial static images (at emission wavelengths around 525 nm, 600 nm, and 700 nm) that were used to identify the properly labeled Ndc80 complexes carrying both donor and acceptor dyes, according to a previously described method (*Deng and Asbury, 2017*). (Because of the method of labeling concurrently with both dyes, the Ndc80 complexes used for single molecule FRET experiments were a mixture of those with one or two donor dyes, one or two acceptor dyes, or one of each at either cysteine residue location.) Records of fluorescence versus time from each camera over the entire duration of the movie were generated by integration of the intensities within 7 × 7 pixel regions of interest centered on the particles. For analysis of Ndc80 complexes in co-complex with GFP-MIND, the three-color particles carrying an end-labeled Ndc80 complex (with both Cy3 and AF647) that colocalized with a GFP-MIND were identified and distinguished from the two-color particles consisting of end-labeled Ndc80 complex alone. For analysis of FRET exhibited by

Ndc80 complexes attached to microtubules, only non-diffusing, two-color Ndc80 complex particles that remained bound to the microtubules over the entire duration of the movie were included.

Before the calculation of FRET levels, raw records of fluorescence versus time were first corrected for background, spillover, and cross-excitation. Background was measured on a slide-to-slide basis and subtracted from the corresponding records.

## Estimation of spillover

While most of the fluorescence emission from the donor Cy3 dyes fell onto our 600 nm camera, their spectrum is broad enough that a fraction of their emission 'spilled over' onto the 700 nm camera. To quantify this spillover effect, we imaged single oligonucleotides labeled with Cy3 alone (under 561 nm excitation) and measured the ratio of their emission at 700 nm divided by their emission at 600 nm,

$$S = \frac{I^{700}_{561}}{I^{600}_{561}}$$

where $S$ represents the spillover ratio, $I^{700}_{561}$ represents the intensity measured at 700 nm emission under 561 nm excitation, and $I^{600}_{561}$ represents the intensity measured at 600 nm emission under 561 nm excitation. For Cy3 molecules in our microscope, $S$ = 0.13 ± 0.01 (mean ± S.D. from N = 63 molecules).

## Estimation of cross-excitation

The 561 nm laser excitation used during our single molecule FRET measurements is near the peak absorption for the donor Cy3 dyes. While this wavelength is considerably shorter than the peak absorption for the acceptor AF647 dyes, it nevertheless directly excites some fluorescence in AF647. To quantify this 'cross-excitation' effect, we imaged single oligonucleotides labeled with AF647 alone and measured the ratio of their emission (at 700 nm) when cross-excited at 561 nm divided by their emission (again at 700 nm) when excited with nominally identical power at 641 nm,

$$X = \frac{I^{700}_{561}}{I^{700}_{641}}$$

where $X$ is the cross-excitation ratio, $I^{700}_{561}$ is the intensity measured at 700 nm emission under 561 nm excitation, and $I^{700}_{641}$ is the intensity measured at 700 nm emission under 641 nm excitation. For AF647 molecules in our microscope, $X$ = 0.37 ± 0.07 (mean ± sdev from N = 46 molecules).

## Correction for spillover and cross-excitation

For FRET measurements with the end-labeled Ndc80 complexes or oligonucleotide controls, intensity versus time at emission wavelengths around 600 nm and 700 nm was measured initially with 561-nm laser illumination (at a nominal power of 25 mW), which after background subtraction yielded records of donor and acceptor emission versus time, $I_D(t)$ and $I^*_A(t)$, respectively. Illumination was then switched to 641 nm (at 25 mW) to measure acceptor intensities at 700 nm emission under direct excitation, $I^{700}_{641}$. After photobleaching the acceptors (with 100 mW at 641 nm), the illumination was switched back to 561 nm (at 25 mW) to measure intensities of the donors alone at 600 nm emission, $I^{600}_{561}$. Each record of acceptor emission was then corrected for spillover and cross-excitation using the equation,

$$I_A(t) = I^*_A(t) - S \cdot I^{600}_{561} - X \cdot I^{700}_{641}$$

where $I_A(t)$ represents the corrected acceptor emission versus time.

## FRET calculation

Single molecule FRET efficiency versus time, $E_{FRET}(t)$, was calculated using the equation,

$$E_{FRET}(t) = \frac{I_A(t)}{I_A(t) + I_D(t)}$$

Histograms of FRET efficiency were generated using concatenated $E_{FRET}(t)$ records from at least 41 individual molecules. The no-FRET peak of the histogram generated from negative control molecules was fit with a Gaussian function to determine a threshold two standard deviations above the center of the Gaussian. Values above this threshold (0.30) were considered high-FRET. To examine the kinetics of switching between no- and high-FRET states, the numbers of upward and downward threshold-crossings in FRET records from 207 individual Ndc80 complexes were counted and the total times spent above and below the threshold were summed. An estimated rate of 'opening', $k_{opening}$, was determined from the total number of downward crossings divided by the total time spent above the threshold. Likewise, an estimated rate of 'closing', $k_{closing}$, was determined from the total number of upward crossings divided by the total time spent below the threshold.

## Estimating how tight bending affects the affinity of Ndc80 complex for microtubules

We envision a four-state reaction scheme in which the closed (tightly bent) and open conformations of the Ndc80 complex are in equilibrium with each other, both on and off the microtubule, as depicted in *Figure 5E*. In order to satisfy the principle of microscopic reversibility, the following relation must hold,

$$K_d^{closed} = K_d^{open}\left(\frac{K_{eq}^{bound}}{K_{eq}^{free}}\right)$$

where $K_d^{closed}$ and $K_d^{open}$ are the equilibrium constants for dissociation from the microtubule by the closed and open conformations, respectively, and where $K_{eq}^{bound}$ and $K_{eq}^{free}$ are equilibrium constants for opening (straightening) of the Ndc80 complex when it is microtubule-bound and free, respectively. Our observation that 6% of microtubule-attached Ndc80 complexes exhibit high FRET implies that $K_{eq}^{bound} \cong 19$. Our observation that 25% of Ndc80 complexes alone exhibit high FRET suggests that $K_{eq}^{free} \cong 3$. (Here, we assume that tethering to the coverslip does not alter the energetics of the opening-closing transition.) Inserting these values into the above equation indicates that $K_d^{closed} \cong 6 \cdot K_d^{open}$, implying a sixfold weaker affinity for microtubules when the Ndc80 complex is tightly bent.

## Acknowledgements

We thank Kimberly Fong, Eric Muller, King Yabut, Amanda Clouser and the Zagotta Lab for technical assistance and advice. We thank Simon Jenni of the Harrison Lab for sharing the Ndc80 complex model. We also thank the members of the TND laboratory, CLA laboratory, and Seattle Mitosis Club for helpful discussions. This work was supported by National Institute of General Medical Sciences Grants T32 GM007270 (to EAS), R01 GM040506 (to TND), R35 GM130293 (to TND), R01 GM079373 (to CLA), and P01 GM105537 (to CLA).

## Additional information

### Funding

| Funder | Grant reference number | Author |
| --- | --- | --- |
| National Institute of General Medical Sciences | T32 GM007270 | Emily Anne Scarborough |
| National Institute of General Medical Sciences | R01 GM040506 | Trisha N Davis |
| National Institute of General Medical Sciences | R35 GM130293 | Trisha N Davis |
| National Institute of General Medical Sciences | R01 GM079373 | Charles L Asbury |
| National Institute of General Medical Sciences | P01 GM105537 | Charles L Asbury |

The funders had no role in study design, data collection and interpretation, or the decision to submit the work for publication.

## Author contributions
Emily Anne Scarborough, Conceptualization, Data curation, Formal analysis, Funding acquisition, Investigation, Writing—original draft, Writing—review and editing; Trisha N Davis, Conceptualization, Supervision, Funding acquisition, Methodology, Project administration, Writing—review and editing; Charles L Asbury, Conceptualization, Software, Formal analysis, Supervision, Funding acquisition, Methodology, Project administration, Writing—review and editing

## Author ORCIDs
Trisha N Davis (iD) http://orcid.org/0000-0003-4797-3152
Charles L Asbury (iD) http://orcid.org/0000-0002-0143-5394

## Decision letter and Author response
Decision letter https://doi.org/10.7554/eLife.44489.028
Author response https://doi.org/10.7554/eLife.44489.029

# Additional files

## Supplementary files
• Transparent reporting form
DOI: https://doi.org/10.7554/eLife.44489.026

## Data availability
All data generated or analysed during this study are included in the manuscript and supporting files. Source data files have been provided for Figures 1, 2, 3, 4 and 5, as well as Figure 1—figure supplement 1, Figure 3—figure supplement 1, Figure 4—figure supplement 1 and Figure 5—figure supplement 1.

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
