## [Decision Letter]

Thank you for submitting your article "Tight bending of the Ndc80 complex provides intrinsic regulation of its binding to microtubules" for consideration by *eLife*. Your article has been reviewed by three reviewers, including Iain M Cheeseman as the guest Reviewing Editor, and the evaluation has been overseen by Anna Akhmanova as the Senior Editor.

The reviewers have discussed the reviews with one another and the Reviewing Editor has drafted this decision to help you prepare a revised submission.

Summary:

The Ndc80 complex is a key microtubule attachment point between microtubules and chromosomes that enables the segregation of chromatids during mitosis. This study employs single-molecule fluorescence microscopy and single molecule FRET to investigate the microtubule-binding properties of the Ndc80 complex. In particular, the authors investigate the role of a large conformational change in regulating microtubule binding by the yeast NDC80 complex. The authors show that NDC80 complex binding to microtubules in vitro is auto-inhibited by two members of the complex, Spc25 and Spc24, and that this auto-inhibition occurs intramolecularly.

The study builds on previous work by the Davis and Asbury (and other) labs that has proposed a model of Ndc80 complex auto-inhibition. This current study demonstrates that, on the single molecule level, the residence time of full-length Ndc80 molecules on microtubules is reduced compared to a Broccoli construct lacking Spc24-25. In their single molecule assays, the authors convincingly show that the NDC80 complex switches dynamically between closed conformations in which its globular ends are close together and open conformations in which its globular ends are far apart. Addition of Spc24-25 in trans reduces the residence time of Broccoli, suggesting an intramolecular inhibition in the context of the full-length complex. The authors further provide evidence that Mis12 binding to Ndc80c prevents a "closed" conformation and thereby relieves the inhibition. Based these results and previous work, the authors conclude that the NDC80 complex exhibits distinct microtubule binding properties in open and closed conformations and that the effect of the MIND complex on NDC80 interactions with microtubules is at least partially mediated by this conformational switch.

Essential Revisions:

The reviewers found the paper to be interesting and a potentially important advance, and were in agreement that this study includes a strong range of biophysical data. The data is of a very high caliber that is a hallmark of this strong collaboration between the Asbury and Davis labs. However, the reviewers identified several things that would be required for a revised version to be acceptable for publication in *eLife*.

First, it would be important to have a more comprehensive range of controls for the biochemical and biophysical experiments, including the use of targeted mutants in selected assays.

1) The use of mutants (Ndc80 d490-510 and Spc25 7K) to disrupt bending and inter-molecular self-association is a powerful part of this paper. However, the use of these mutants is at present incomplete, preventing the strongest statements possible in many of their experiments. These two mutants should be used more systematically across the assays in the paper.

- Spc25 7K is used when the Spc24/25 dimer is added in trans to Ndc80 Broccoli (Figure 1), but is not used in the context of the intact complex and is also used primarily for microtubule binding assays. It is well established that the positively charged N-terminal tail is important to Ndc80 binding in vitro, so the specificity of adding a highly negatively charged protein (Spc24/25) as a competitor is unclear. This concern and many others would be eliminated by performing the measurements with a full-length complex containing the 7K point mutations on Spc24/25. Assuming this mutant has a longer residence time, this would demonstrate that the Spc24/25 can inhibit microtubule binding in cis. Moreover, if the MIND complex still binds to this version of the Ndc80 complex (which should be tested using in vitro binding – for example, by gel filtration), then there should be no additional effect of MIND addition if MIND stimulation acts by relieving the auto-inhibition.

- The Ndc80 d490-510 is used as a control in the single molecule FRET experiments to reduce bending, but is not used for the bulk FRET experiments (Figure 2) or microtubule binding experiments. Based on their model, these mutants should at least partially mimic the addition of the MIND complex. It would be very helpful to include this mutant in the FRET experiments and other assays wherever possible. In addition, the d490-510 mutant would help to test the possibility of inter-molecular interactions in Figure 2.

2) It would be good to show that it is the physiological mode of Mis12c binding to Ndc80c that is important for the relief of inhibition and the increase in residence time of full-length Ndc80c. Based on the existing data, the effect of Mis12c could either be counteracted by Spc24-25 binding to the Ndc80 head (as depicted in the illustration), or by Spc24-25 titrating away Mis12c from full length Ndc80c. For this, either a mutant Ndc80c complex (point mutations in Spc24-25) or a mutant Mis12c that abolishes binding between Ndc80c-Mis12c could be used. As the Davis lab has characterized these mutations before, it should be feasible to include them in these in vitro assays.

Second, although the reviewers appreciate that this is a primarily biophysical study, complementary support from the analysis of mutants in vivo, if possible, would significantly strengthen this paper by allowing the authors to justify the physiological relevance of their findings. This is a key advantage of using the budding yeast proteins that the ability to test selected mutants using replacement assays should be feasible. This is particularly true of the mutants that were newly generated for this paper, and the reviewers recommend testing the Spc25 7K mutant in cells. The reviewers recognize that the initial paper did not include such genetic approaches, but we believe the ability to test these mutants in cells would increase the broader impact of this paper. If this mutant has additional consequences beyond disrupting the intra-molecular Ndc80 complex interaction (for example, disrupting MIND complex binding), it would be important to indicate this for considering the results from the biochemical assays and their overall model.

---

## [Author Response]

Essential Revisions:The reviewers found the paper to be interesting and a potentially important advance, and were in agreement that this study includes a strong range of biophysical data. The data is of a very high caliber that is a hallmark of this strong collaboration between the Asbury and Davis labs. However, the reviewers identified several things that would be required for a revised version to be acceptable for publication in eLife.

We thank the reviewers for their careful consideration of our manuscript, and their generally positive view of our work.

First, it would be important to have a more comprehensive range of controls for the biochemical and biophysical experiments, including the use of targeted mutants in selected assays.1) The use of mutants (Ndc80 d490-510 and Spc25 7K) to disrupt bending and inter-molecular self-association is a powerful part of this paper. However, the use of these mutants is at present incomplete, preventing the strongest statements possible in many of their experiments. These two mutants should be used more systematically across the assays in the paper.- Spc25 7K is used when the Spc24/25 dimer is added in trans to Ndc80 Broccoli (Figure 1), but is not used in the context of the intact complex and is also used primarily for microtubule binding assays. It is well established that the positively charged N-terminal tail is important to Ndc80 binding in vitro, so the specificity of adding a highly negatively charged protein (Spc24/25) as a competitor is unclear. This concern and many others would be eliminated by performing the measurements with a full-length complex containing the 7K point mutations on Spc24/25. Assuming this mutant has a longer residence time, this would demonstrate that the Spc24/25 can inhibit microtubule binding in cis. Moreover, if the MIND complex still binds to this version of the Ndc80 complex (which should be tested using in vitro binding – for example, by gel filtration), then there should be no additional effect of MIND addition if MIND stimulation acts by relieving the auto-inhibition.

We would have very much liked to examine the effects of the 7K charge-reversal mutant in the context of a full-length heterotetrameric Ndc80 complex. However, we have made three independent attempts to purify this mutant complex and, unfortunately, all three failed to produce a stable heterotetramer. (In our revised manuscript, these failures are noted in the Materials and methods section.) For this reason, we have limited our use of the 7K mutant to experiments with only the 7K Spc24/Spc25 dimer, which is stable and biochemically well-behaved.

We also agree that it is important to understand whether Spc24/Spc25 can inhibit microtubule binding in cis. We believe our data already address this point directly: Our microtubule-binding assays were performed at very low concentrations of Ndc80 complex, where the decoration of microtubules was very sparse, allowing individual complexes to be discerned. Under these conditions, inter-complex interactions are negligible and the shorter residence time of the full-length wild-type heterotetramer relative to Broccoli (Figure 1B and 1C) therefore indicates that the inhibition must occur within individual complexes.

- The Ndc80 d490-510 is used as a control in the single molecule FRET experiments to reduce bending, but is not used for the bulk FRET experiments (Figure 2) or microtubule binding experiments. Based on their model, these mutants should at least partially mimic the addition of the MIND complex. It would be very helpful to include this mutant in the FRET experiments and other assays wherever possible. In addition, the d490-510 mutant would help to test the possibility of inter-molecular interactions in Figure 2.

We thank the reviewers for suggesting additional experiments using the Ndc80 Δ490-510 partial loop deletion mutant. For our revised manuscript, we have collected new data showing that, indeed, the mutant Ndc80 Δ490-510 complex binds better to microtubules than wild-type (Figure 4E and 4F), partially mimicking the addition of MIND, as predicted. We respectfully disagree that testing Ndc80 Δ490-510 under bulk FRET conditions would be additionally informative. Our single-molecule FRET data (Figure 4C and 4D) already indicate that Ndc80 Δ490-510 exhibits lower FRET than wild-type and provide the distribution of FRET across a population of molecules. Bulk FRET would be redundant and would only report an average FRET value for the population.

2) It would be good to show that it is the physiological mode of Mis12c binding to Ndc80c that is important for the relief of inhibition and the increase in residence time of full-length Ndc80c. Based on the existing data, the effect of Mis12c could either be counteracted by Spc24-25 binding to the Ndc80 head (as depicted in the illustration), or by Spc24-25 titrating away Mis12c from full length Ndc80c. For this, either a mutant Ndc80c complex (point mutations in Spc24-25) or a mutant Mis12c that abolishes binding between Ndc80c-Mis12c could be used. As the Davis lab has characterized these mutations before, it should be feasible to include them in these in vitro assays.

We are grateful to the reviewers for this excellent suggestion. We agree that the decrease in residence time of the Ndc80-MIND co-complex upon addition of excess Spc24/Spc25 could have resulted from a titration of MIND away from the Ndc80 complex. To test this possibility, we have purified a mutant Spc24/Spc25 dimer carrying a V159D point mutation in the Spc25 protein (as shown in Figure 1—figure supplement 3A). Previous work by the Westermann group (Malvezzi et al., 2013) established that this mutant cannot bind the MIND complex (a result which we confirmed; Figure 1—figure supplement 3B). When added in trans, the mutant V159D Spc24/Spc25 dimer has the same inhibitory effect as wild-type Spc24/Spc25 (Figure 1—figure supplements 3C and 3D). This indicates that the Spc24/Spc25 dimer interacts directly with the Ndc80 complex for inhibition, rather than competing away MIND (as noted in the revised manuscript, in subsection “Microtubule binding by the Ndc80 complex is auto-inhibited by its Spc24/Spc25 end”).

Second, although the reviewers appreciate that this is a primarily biophysical study, complementary support from the analysis of mutants in vivo, if possible, would significantly strengthen this paper by allowing the authors to justify the physiological relevance of their findings. This is a key advantage of using the budding yeast proteins that the ability to test selected mutants using replacement assays should be feasible. This is particularly true of the mutants that were newly generated for this paper, and the reviewers recommend testing the Spc25 7K mutant in cells. The reviewers recognize that the initial paper did not include such genetic approaches, but we believe the ability to test these mutants in cells would increase the broader impact of this paper. If this mutant has additional consequences beyond disrupting the intra-molecular Ndc80 complex interaction (for example, disrupting MIND complex binding), it would be important to indicate this for considering the results from the biochemical assays and their overall model.

We agree that complementary analyses of mutants in vivo are critical for establishing the physiological relevance of our in vitro findings. In our view, two prior in vivo studies already provide strong justification but, in retrospect, our original manuscript failed to explain this clearly enough. We have added a more comprehensive discussion to the revised manuscript (Discussion section) and also summarize the key points here: Published work from the Tanaka lab (Maure et al., 2011) has shown that Ndc80 Δ490-510, lacking most of the loop region, causes temperature-sensitive cell growth and frequent failures in biorientation of sister kinetochores. Likewise, our own published study has shown that a pair of point mutations near the Ndc80 loop causes temperature-sensitive cell growth and defects in kinetochore biorientation (Tien et al., 2014). The in vivo phenotypes caused by these loop-proximal mutations can be suppressed by additional cis-acting mutations made on exactly the opposite side of the loop, which represents strong genetic evidence for a tightly bent conformation of the Ndc80 complex (Tien et al., 2014), and was the original motivation for our current in vitro study. Moreover, the same pair of loop-proximal point mutations enhances microtubule-binding of the Ndc80 complex in a manner that mimics the addition of MIND (Kudalkar et al., 2015). Now our new findings suggest that all these prior observations could have a single mechanistic basis – i.e., that tight bending of the Ndc80 complex might be required for proper regulation of kinetochore-microtubule attachments in vivo.

Unfortunately, as explained above and noted in subsection “Protein expression and purification” of the revised manuscript, the 7K charge-reversal mutant appears to destabilize formation of the heterotetrameric Ndc80 complex, which would complicate interpretation of any associated phenotypes, since they could arise simply from a disruption of the entire Ndc80 complex, rather than specifically from inhibition of tight folding. For this reason, we have not analyzed the 7K mutant in vivo.